# Trials of a Fluorescent Endoscopic Video System for Diagnosis and Treatment of the Head and Neck Cancer

**DOI:** 10.3390/jcm8122229

**Published:** 2019-12-17

**Authors:** Dina Farrakhova, Artem Shiryaev, Dmitry Yakovlev, Kanamat Efendiev, Yulia Maklygina, Alexandr Borodkin, Maxim Loschenov, Lina Bezdetnaya, Anastasia Ryabova, Liana Amirkhanova, Svetlana Samoylova, Mikhail Rusakov, Victor Zavodnov, Vladimir Levkin, Igor Reshetov, Victor Loschenov

**Affiliations:** 1Prokhorov General Physics Institute of the Russian Academy of Sciences, 119991 Moscow, Russia; us.samsonova@physics.msu.ru (Y.M.); borodkin.aleksandr@gmail.com (A.B.); maxvl2000@gmail.com (M.L.); nastya.ryabova@gmail.com (A.R.); loschenov@mail.ru (V.L.); 2University Clinical Hospital no. 1, Oncology Center, I.M. Sechenov First Moscow State Medical University (Sechenov University), Ministry of Health of the Russian Federation, 119991 Moscow, Russia; artemdoc@mail.ru (A.S.); liana_1994@mail.ru (L.A.); sv_samoilova75@mail.ru (S.S.); m1rus@mail.ru (M.R.); marina-zavodnova@rambler.ru (V.Z.); doctor-levkin@mail.ru (V.L.); reshetoviv@mail.ru (I.R.); 3Shemyakin and Ovchinnikov Institute of Bioorganic Chemistry of the Russian Academy of Sciences, 117997 Saratov, Russia; yakovlevrsmu@gmail.com; 4Department of Laser Micro-, Nano-, and Biotechnology, Institute of Engineering Physics for Biomedicine, National Research Nuclear University “MEPhI”, 115409 Moscow, Russia; kanamatius@mail.ru; 5Centre de Recherche en Automatique de Nancy, CNRS, Université de Lorraine, 54519 Vandœuvre-lès-Nancy, France; l.bolotine@nancy.unicancer.fr; 6Institut de Cancérologie de Lorraine, 54519 Vandoeuvre-lès-Nancy, France

**Keywords:** fluorescent diagnostics, photodynamic therapy, fluorescent endoscopic video system, fluorescent intraoperative navigation, photosensitizers, chlorin e6, oncological diseases, head and neck cancer

## Abstract

This article presents the results of intraoperative fluorescent diagnostics via the endoscopic system for assessing the quality of photodynamic therapy (PDT) of head and neck cancer. The diagnosis and PDT procedures were performed on the five patients with malignant neoplasms of the vocal cords, lateral surface of the tongue, and trachea and cancer of the left parotid salivary gland. Molecular form of chlorin E6 (Ce6) was intravenously administered with a 1.0–1.1 mg/kg concentration for PDT. Fluorescent diagnostics (FD) was conducted before PDT and after PDT procedures. Control of PDT efficiency was carried out by evaluating the photobleaching of the drug (photosensitizer). The method of intraoperative fluorescent imaging allows determining the exact location of the tumor and its boundaries. The assessment of photosensitizer photobleaching in real time regime allows making quick decisions during PDT procedure, which helps improving the quality of patients’ treatment. The results showed the convenience of endoscopic fluorescent video system in various nosologies of head and neck cancer. Therefore, this diagnostic approach will improve the effectiveness of cancer treatment.

## 1. Introduction

Nowadays, more than half a million people in the world are diagnosed with a head and neck cancer [1,2,3]. Head and neck cancer is a heterogeneous group of tumors and includes malignant neoplasms of various histological structures localized on the lip mucous membrane, mouth, pharynx, larynx, cervical esophagus, nasal cavity, paranasal sinuses, and salivary glands [4]. Head and neck cancer in its late stages can spread gradually to other parts of the body. Also, tumor cells can spread through the lymphatic flow, nervous system, and blood vessels to other body systems. The traditional treatment of leukoplakia, erythroplakia, erythroleukoplakia, hyperplasia, and submucosal fibrosis is a complete surgical resection, which leads to the formation of scar tissue. The complex anatomy of mucous membrane in the head and neck area requires an extensive tissue resection, which impairs the quality of patients’ life, resulting in serious functional and aesthetic deficiencies, such as chewing, swallowing, and speaking ability [5].

Another problem of treatment of head and neck cancer is the inaccurate determination of tumor boundaries. The cause of it is incomplete resection of cancer cells, resulting in a relapse of the disease. Therefore, determination of the tumor boundaries with a high accuracy degree is a crucial factor to maximize the effectiveness of treatment and the subsequent quality of patients’ life [5]. At the first stage of diagnostics, doctors use methods such as visual examination and palpation for differentiating tumor tissue from healthy one [6]. These methods are subjective and do not quantify the diagnosis of pathological neoplasms. However, the more exact methods as radio frequency spectroscopy, Raman spectroscopy, photoacoustics, and optical coherence tomography are beginning to be applied more and more in oncology. All of them have an inadequate choice, a low risk of procedural errors, and a need for an objective interpretation of the results [7,8,9,10].

The intraoperative fluorescence imaging provides reliable information about the borders of the resected tumor. This method ensures a quantitative approach of the analysis of tissue type during surgery by accumulation of exogenous fluorophores in cancer cells. Selective accumulation of photosensitizers (PS) in malignant cells makes possible to identify the boundaries and extent of cancerous lesions by the fluorescence characteristics. Subsequent laser irradiation with sufficient power and appropriate wavelength corresponding to PS absorption peak leads to the cytotoxic effect of cancer cells. This method is called photodynamic therapy (PDT). Consequently, intraoperative fluorescence diagnostics can reduce the risk of relapse and metastasis due to the effectiveness of determination of the tumor boundaries. The main advantage of this method is the ability to determine the boundaries of the cancer tissue in real time with a short period. Fluorescence excitation is performed in the red spectrum range, which corresponds to the “transparency window” of biological tissue. In this area, the light is less scattered on the components of biological tissues, which allows light to penetrate deeper into biological tissue and increasing the depth of probe [11]. The light sources of diagnostic video system operate in a continuous mode. It allows visualizing images of the pathological tissue with natural colors and separating graphic layer with fluorescence. The visualization of fluorescence map on the top of a color image on the monitor generally improves the sensitivity of optical fluorescence intraoperative navigation during malignant tumors removing of head and neck.

Thus, intraoperative fluorescence diagnostics increases the efficiency of PDT by continuously and simultaneously monitoring the concentration of PS in the irradiated area. Also, PDT, as a local method of exposure, has certain advantages and can be used repeatedly and can increase the overall survival of patients with malignant neoplasms.

## 2. Experimental Section

The endoscopic video system (Biospec LLC, Moscow, Russia) was used for fluorescent diagnosis (FD). The system consists of a white light source, a 635 nm red laser for Ce6 excitation, and color and black-and-white cameras for imaging tumor tissue. The flexible and rigid endoscopes were used as a delivery system for FD (Figure 1).

The images in visible and fluorescent light in the areas of interest were obtained via video fluorescent system. In this case, the ultimate conclusion on the tumor process can’t be derived based on the image in visible light. Video fluorescence staining significantly increases the accuracy of visualization. Besides, a diagnostic marker is installed on the image fragment of the studied area allowing to measure the fluorescence on the selected area in real time. For determining the fluorescence index, the software of the video fluorescence system calculates the average pixel intensity of the image obtained from the monochrome video camera in the highlighted area by the diagnostic cursor and normalizes it to the parameters of the monochrome video camera. Further, the average pixel intensity of the image of the color video camera’s red channel in the area highlighted by the diagnostic cursor is calculated and normalized to the parameters of the color video camera. Then, the first obtained value is normalized to the second one, thus obtaining the relative concentration of PS in the pathological tissue. The value of the normal tissue was determined by fluorescence and the degree of PS accumulation at two different points in the unchanged part of the studied organ. Identical values were obtained in these two positions, allowing to be considered as a normal reference value [12].

PDT was conducted by the therapeutic laser with 660 nm generation wavelength at the maximum of Ce6 accumulation in the patient’s body. The fluence of PDT was varied from 20 to 100 J/cm^2^ depending on the location of tumor and the pain feelings of patients. The power density was 40–100 mW/cm^2^ at the output fiber tip. The areas with malignant neoplasms in the vocal cords, trachea, root of the tongue, and the oropharynx were irrigated by lidocaine solution for reducing pain and further gentle therapy.

A cylindrical diffuser designed for interstitial therapy and endoscopic operations was used as a system for light delivery to a tumor in the oropharynx region radiation in all directions (Figure 2a). Optical fiber with direct output radiation was used for operations on a patient’s tissue surface. A scattering lens was established at the tip of the light fiber for a homogeneous distribution of the light spot (Figure 2b).

Visualization of cell nuclei was performed by Acridine Orange (AO) staining of fresh biopsy material 4 h after biopsy sampling. The AO stained samples were transferred to a Petri dish with a thin glass bottom thickness 0.16 mm in PBS. The fluorescence distribution in the samples was recorded using an LSM-710 laser scanning confocal microscope (Carl Zeiss Microscopy, Jena, Germany). The EC Plan-Neofluar 10×/0.3 objective was used. The fluorescence of AO was recorded in the wavelength range 500–600 nm, upon laser light excitation with a wavelength 488 nm. The Ce6 was recorded in the wavelength range 650–720 nm (λexc 633 nm). As a result, the distribution of AO staining (green pseudo-color in images) and Ce6 (red pseudo-color in images) were overlaid on top of images obtained in transmitted light mode. The three-dimensional images were obtained by registering a series of images with a step of 5 μm along the Z-axis, followed by 3D image reconstruction.

Ce6-based PS was used for FD as a fluorescent agent for the navigation and control of the efficiency of PDT by photobleaching. Also, Ce6 was used as a PS for the malignant neoplasms’ treatment. Ce6 is a well-known PS in molecular form, which is used in clinical trials with the commercial name Photoditazine (Veta-Grand LLC, Moscow, Russia) and it is a concentrate for the preparation of an infusion solution [13,14,15]. The PS absorption spectrum has an intense peak at 661–662 nm wavelength (Figure 3b).

For i.v. administration, a sterile Ce6 aqueous solution was prepared at the concentration 1.0–1.1 mg/kg. The maximum accumulation of PS was observed 2 h after intravenous administration. FD of the pathological site was performed before and immediately after PDT procedure. The PS has an affinity to the tumor type, which was considered in this study. The drug accumulates sufficiently in the tumor tissue, which allows identifying the difference in the fluorescent signal in healthy tissue in comparison with pathological tissue. In addition, the drug accumulation depends on the time of diagnosis after PS administration.

Daylight avoidance regime is necessary to keep for 2–3 days after the PDT procedure. No cases of skin phototoxicity have been reported and there were no postoperative complications of patient’s therapy. The treatment of different tumor localizations of the head and neck was monitored 2–3 months after the PDT procedure.

The various nosologies were selected with different location and depth of invasion and the process stage for assessing the effectiveness of PDT control via the video endoscopic fluorescent system. The study involved 5 patients diagnosed with malignant neoplasms of the right vocal cord, trachea, lateral surface of the tongue, left parotid salivary gland, and left vocal cord (2019 ICD-10-CM). The age of patients ranged from 58 to 94 years. Diagnoses were confirmed by biopsy, which was obtained before PDT.

## 3. Results

All patients demonstrated the high fluorescence intensity in the area of the malignant neoplasm in comparison with healthy tissue in 2 h after the PS injection. The fluorescent contrast of exogenous fluorophores indicates the selectivity of drug accumulation in pathological foci. The patients felt some discomfort, such as a burning sensation in the area of radiation or pain reactions during the therapy. Lidocaine irrigation of the mucous tissue in the area of pathology contributed to sufficient analgesia for further therapy. PDT on the vocal cords was performed with 22 J/cm^2^ fluence for the comfortable therapy process. Anamnesis records indicated that 40 years ago this patient had already undergone the removal of benign tumors of the vocal cords. The patient had complaints on hoarseness of voice and dry cough in March 2019. The diagnosis revealed multiple formations on the vocal cords and squamous cell cancer was detected after histological research. Considering the patient’s age (94 years old) and the concomitant disease, the Oncological Council recommended to conduct PDT. Evaluation of the treatment’s effectiveness in the patient with malignant neoplasm of the vocal cords was performed 7 days after the PDT procedure. The results showed the tumor regression after the first PDT session (Figure 4). The repeated diagnosis of the patient vocal cords was performed in 1.5 months, which did not reveal a tumor recurrence.

A biopsy was taken in a pathological area with high fluorescence intensity. The intracellular distribution of PS in cancer cells was investigated by confocal microscopy (Figure 5).

A large area of the cell’s cluster was selected for biopsy analysis. Figure 5 shows two different areas of biopsy material with maximum drug accumulation. Ce6 is not distributed evenly in the tumor tissue due to the heterogeneity of the neoplasm, but in single cell there is a uniform distribution of PS across the cytoplasm.

A pathohistological analysis of the biological tissue taken from the tumor site of the right vocal cord is displayed on the Figure 6 and Figure 7.

A histological analysis showed squamous cell carcinoma, G2, before PDT (Figure 6). The microscopic analysis of biopsy sample demonstrated changes in the tumor cells with paretic plethora of the blood vessels of the microvasculature with the formation of blood clots after PDT (Figure 7a,b). There is a positive dynamic observation in the form of complete regression of the tumor.

Three PDT procedures were carried out on a patient diagnosed with IV stage of the left parotid salivary gland cancer previously subjected to surgical treatment, radiation, and chemotherapy (Figure 8). There are metastatic damages of the soft tissues of the left half of the face, neck, and supraclavicular region and the defeat of the skeleton bones. The histological research indicated mucoepidermal carcinoma of the left salivary gland with perivascular, perineural, and intraneural growth. The disease was progressing and an extended cervical lymphadenectomy was performed on the left part in November 2018. The infiltration in the area of the previously performed operation was noted and histological research showed adenocarcinoma metatstasis in February 2019. At that time, the metastasis removal of the left lateral surface of the neck was performed. In the postoperative period, the Oncological Council recommended several PDT treatments. Two PDT sessions were conducted with a 1 week interval. FD was conducted before and after PDT for assessing changes in the fluorescence marker at the tumor tissue.

The fluorescence intensity gradient was recorded before and after therapy. The fluence of the first PDT session was 70 J/cm^2^. The patient had a burning painful sensation during the exposure. The change in the fluorescence index was evaluated via a combined image generated by the video system software. Metastases are localized intradermally, as was observed during the FD. An area with metastases was irradiated from the front of the patient. Some areas on the skin surface were selected for assessment of the PS photobleaching (Figure 9).

A visual reduction of the Ce6 fluorescence intensity was observed in the fluorescent light mode after PDT. The combined image recorded a decrease in the PS fluorescence 2.8 times, which indicates the photobleaching of the drug. The second PDT operation was performed in a week after the first session. The fluence of the same pathological sites was 80 J/cm^2^ (Figure 10).

Figure 10 shows combined images of the tumor obtained by color and fluorescence cameras. The visualization demonstrates a clear difference between the tumor and normal tissue with a double intense fluorescent signal. The fluorescent signal decreased 2.2 times after PDT.

The visual decrease in edema of the left front side is observed in 2.5 months after PDT (Figure 11). The patient noted a significant improvement in his condition and quality of life. FD was conducted with Ce6, which showed a weak accumulation of the drug in the pathological tissue, thus, indicating a decrease of the tumor size. The results of MRI and PET CT refute the presence of tumor recurrence at present time.

PDT of a malignant neoplasm of the lateral surface of the tongue was carried out with 100 J/cm^2^ fluence. The patient was diagnosed with squamous cell carcinoma of the tongue root on the right part (T2N0M0). The patient refused a proposed surgical treatment accepting the treatment of malignant neoplasm by radiation therapy. However, there was a continuous growth of the tumor 12 months after treatment. The patient agreed to have a combination treatment: neoadjuvant PDT followed by surgical treatment. The fiber was introduced through the nasopharynx for PDT procedure (Figure 12).

Figure 13 shows combined images of the surface of the tongue back in three different areas. The value of the fluorescence index decreased 1.9–3.5 times in different parts of the lateral tongue surface after PDT. The images show photobleaching of Ce6, which indicates positive tumor cell destruction. A hemiglossectomy was performed 6 days after PDT. Also, a histological research revealed therapeutic pathomorphism of the first degree. The repeated FD and PDT were performed in a month after the first one (Figure 14).

The FD before the second PDT session showed less fluorescence signal in comparison with the fluorescence signal obtained before surgery resection. PDT was conducted with 100 J/cm^2^ fluence. The FD showed an almost complete photobleaching of the PS in the tumor. Indeed, we noted a 6 times location for the reason that the decrease of the Ce6 fluorescent signal. Today, no data about relapse have been obtained under dynamic observation.

Figure 15 shows a photo of the treatment area of the patient diagnosed with cancer of the lateral tongue surface 2 days after the PDT session. The formation of necrotic tissue was seen on the mucous membrane.

The PDT treatment was carried out on a patient diagnosed with malignant neoplasm of the trachea (T4N1M0) germination into the mediastinum and complicated by stenosis. The patient had 5 courses of polychemotherapy (paclitaxel + carboplatin) before PDT. A nitinol coated stent was installed in the area of tracheal stenosis under endoscopic control before the treatment according to urgent indications. Intraluminally, PDT presents a high risk of trauma according to the coated stent. Therefore, it was decided to conduct an interstitial PDT procedure. Tracheal cancer PDT was monitored by a spectroscopic system with a modified stereotactic cannula [16]. The introduction of the stereotactic cannula was performed by percutaneous puncture and controlled by ultrasound during tissue resection (Figure 16).

The spectroscopic analysis shows a change in the fluorescence intensity in the pathological zone. Ce6 accumulation was not observed outside the pathological zone. The fluorescence was recorded in the 650–690 nm range with a pronounced peak at 670 nm, which corresponds to Ce6 (Figure 17a). The decrease of the fluorescence signal intensity was registered after the PDT session. The bar plot show a 1.7-fold decrease in Ce6 fluorescence (Figure 17b). The absence of PS in normal tissue is confirmed by the absence of a pronounced fluorescent signal.

A biopsy specimen was taken from the trachea during the diagnosis for studying the intracellular accumulation of PS after PDT using a stereotactic cannula (Figure 18).

The Ce6 fluorescence is still observed on microscopic images after PDT. This confirms the insufficient fluence of irradiation for complete PS photobleaching. On the cellular level, the distribution of Ce6 is diffuse in the cytoplasm of cancer cells.

The patient with a malignant neoplasm of the trachea with spread to the soft tissues was examined in a month after the PDT procedure. The pathological formation was no more visible or palpable. The area of introduction of the stereotactic biopsy fiber is circled in Figure 19 and demonstrates a good cosmetic effect. At the moment, the patient is under dynamic observation, the condition is stable, and there are no signs of tumor progression.

The next clinical observation was an 82-year-old patient with squamous cell carcinoma of the left vocal cord (Figure 20). PS was introduced at a 0.9 mg/kg concentration and PDT fluence was 50 J/cm^2^.

The formation of necrotic tissue and plaque of fibrin was observed 7 days after PDT (Figure 21). These factors indicate the destruction of cancer cells. A complete regression of the tumor was further observed. Further observation of the patient was impossible due to his death from a massive stroke.

The spectroscopy research of pathological part after PDT procedure showed the reduction of Ce6 fluorescent signal. The decrease of Ce6 fluorescent signal after PDT within the lesion area is explained by the partial destruction of PS molecules and a partial PS transformation into other forms. An exact value of the PS’s fluorescence intensity decrease cannot be calculated, because this value depends on the content of oxygen molecules and the PS localization in tumor cells (membrane, cytoplasm, and organelles).

## 4. Discussion

In this study, the assessment of the PDT’s effectiveness of the head and neck malignant diseases was carried out by Ce6 photobleaching upon laser irradiation. The endoscopic fluorescent system allowed obtaining the images of the studied tissue in hard-to-reach places. FD for the assessment of the location of malignant tumors was carried out at the maximum accumulation of Ce6. A zone without pathology where no drug accumulation was visible by fluorescence was selected as normal tissue. The fluorescence of Ce6 was recorded in the 640–700 nm range, which includes the peak fluorescence of the drug. The drop of Ce6 concentration on the disease site to the values of normal tissue or slightly above normal was observed after therapy. According to the previous experience of clinical studies, it was found that the absence of fluorescence after the photodynamic treatment indicates a thrombosis of the tumor vascular network. This phenomenon is a good prognosis reflecting the quality of the therapy session. The location and nature of the tumor formation included in this studied group were based on several criteria: (1) inconvenient location of the tumor for standard surgical treatment, which would require radical surgery, significantly reducing the quality of patient life; (2) a limited number of surgical equipment thus preventing an effective treatment; (3) the high recurrence rates.

The possibility of early diagnosis is a crucial factor for reduction of the high mortality rate of cancer diseases. The video system showed good results by the visualization of malignant tumors of the right side of the vocal cords, trachea, and lateral surface of the tongue and metastases of the parotid gland. The obtained results indicate that the use of PS with a corresponding fluence could be sufficient for the selective accumulation of the drug in pathological foci, which can be used for intraoperative exposure of PDT. The control of Ce6 in cancer cells allows evaluating the effectiveness of therapy intraoperatively by photobleaching immediately after PDT. Thus, oncologists enable immediately to make decisions for the successful treatment of head and neck cancer. Therefore, intraoperative fluorescence diagnostics allows determining a more precise location and boundaries of the tumor, increasing the efficiency of PDT, favorably affecting the survival median of patients with head and neck cancer without compromising of the quality of life.

## Figures and Tables

**Figure 1 jcm-08-02229-f001:**
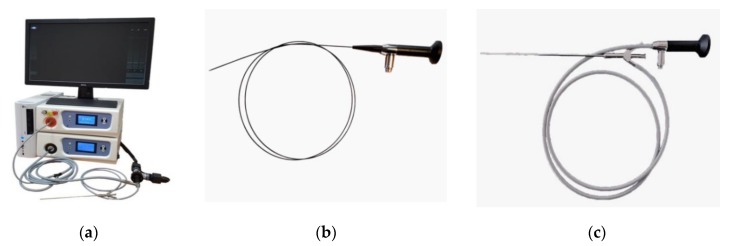
Images of (**a**) endoscopic fluorescent system; (**b**) flexible endoscope; (**c**) rigid endoscope.

**Figure 2 jcm-08-02229-f002:**
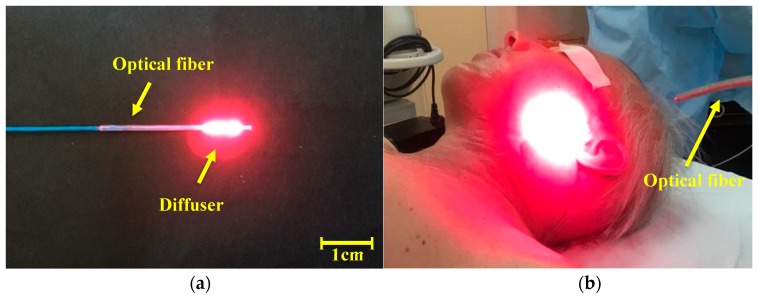
The images of the light optic fiber, which were used in photodynamic therapy (PDT): (**a**) the cylindrical diffuser fiber; (**b**) the direct output radiation optical fiber with scattering lens.

**Figure 3 jcm-08-02229-f003:**
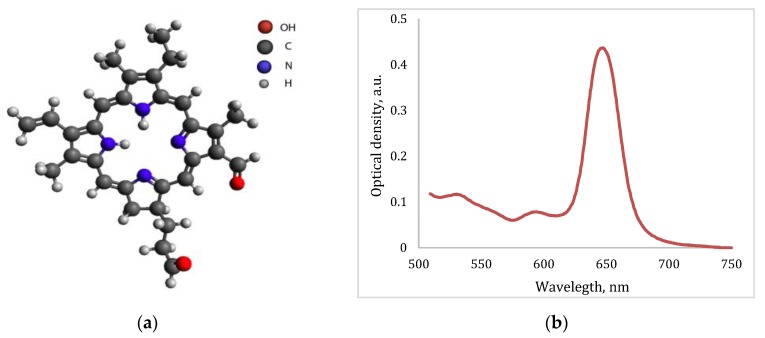
(**a**) The chemical structure of Ce6; (**b**) the absorption spectrum of Ce6.

**Figure 4 jcm-08-02229-f004:**
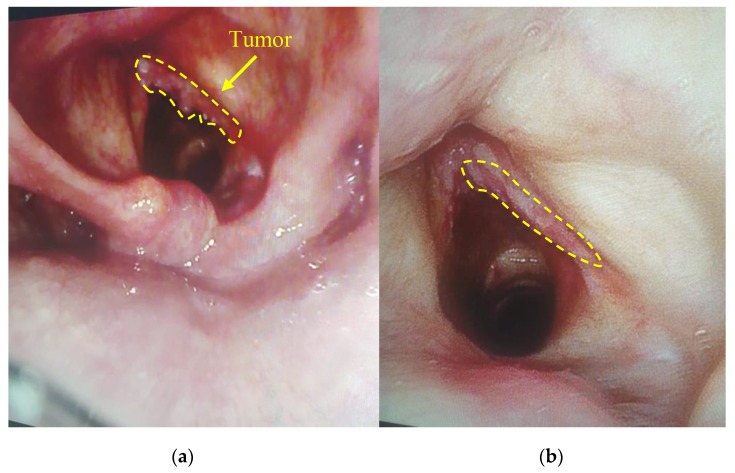
The visible images of vocal cord: (**a**) before PDT; (**b**) in week after PDT.

**Figure 5 jcm-08-02229-f005:**
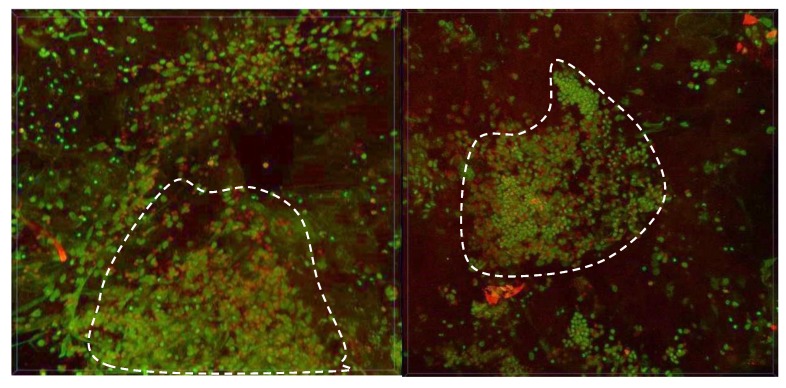
The 3D reconstruction of fluorescent images. Green color corresponds to cells nuclei and red color corresponds to Ce6 fluorescence.

**Figure 6 jcm-08-02229-f006:**
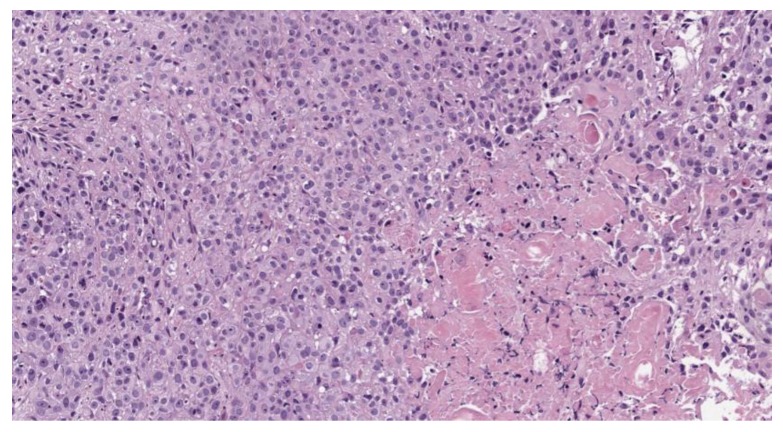
Moderately differentiated squamous cell carcinoma, G2; staining with hematoxylin-eosin, *100 magnification.

**Figure 7 jcm-08-02229-f007:**
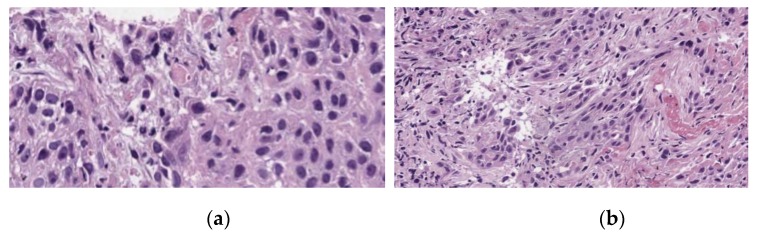
Moderately differentiated squamous cell carcinoma, G2, after PDT; staining with hematoxylin-eosin: (**a**) *400 magnification; (**b**) *100 magnification.

**Figure 8 jcm-08-02229-f008:**
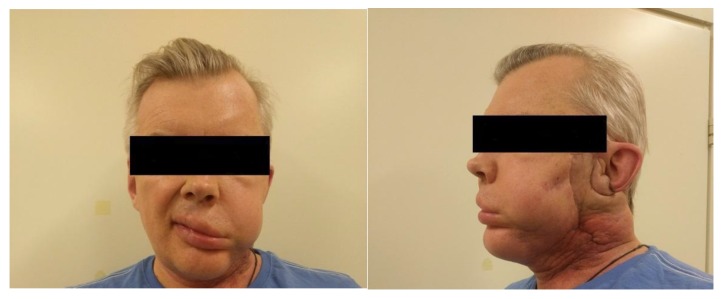
The photo of patient diagnosed with metastasis of parotid gland.

**Figure 9 jcm-08-02229-f009:**
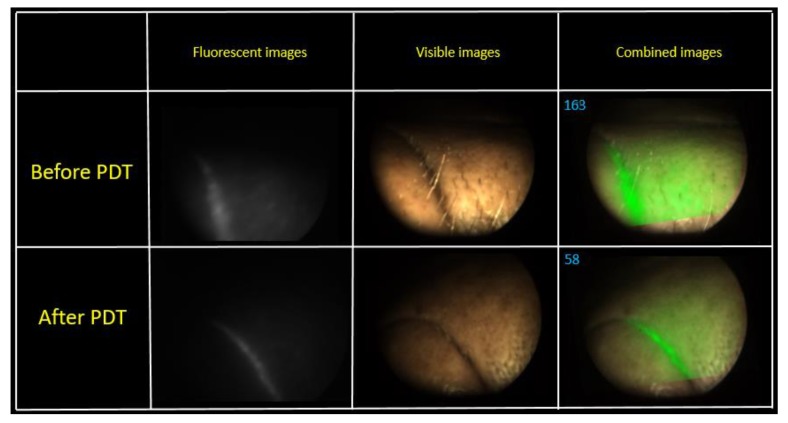
The combined images of tumor area at the first PDT session.

**Figure 10 jcm-08-02229-f010:**
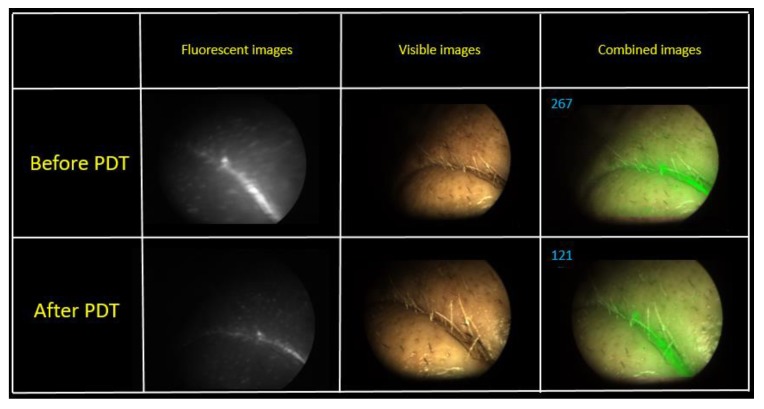
The combined images of tumor area at the second PDT session.

**Figure 11 jcm-08-02229-f011:**
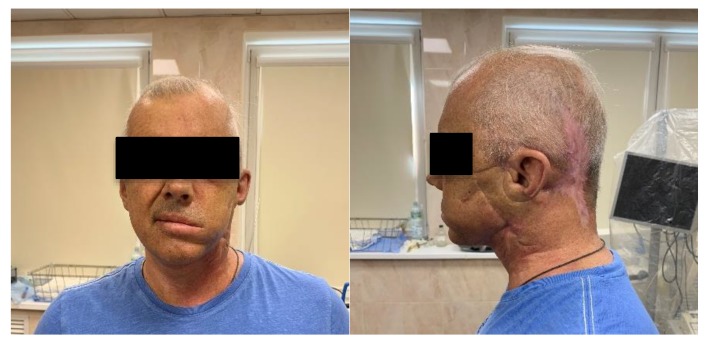
The photo of patient diagnosed with cancer of left parotid salivary gland.

**Figure 12 jcm-08-02229-f012:**
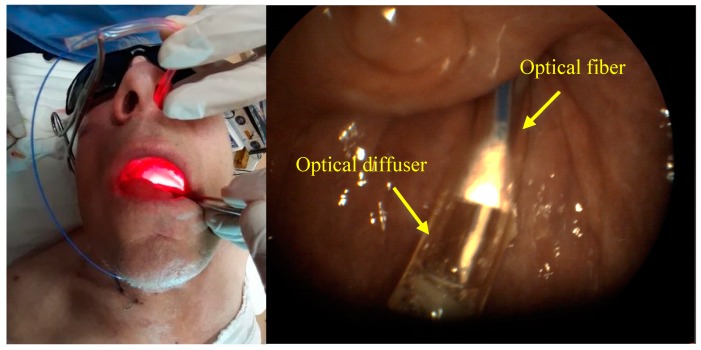
Input of cylindrical diffuser (optical fiber) into the patient nasopharynx diagnosed with malignant neoplasm of the lateral tongue surface.

**Figure 13 jcm-08-02229-f013:**
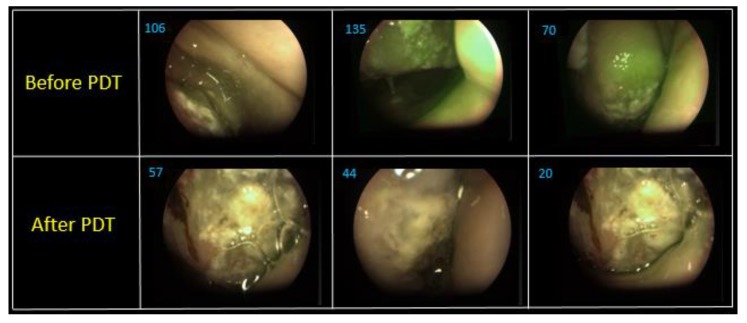
The combined images of mucous and tongue surfaces of three various areas.

**Figure 14 jcm-08-02229-f014:**
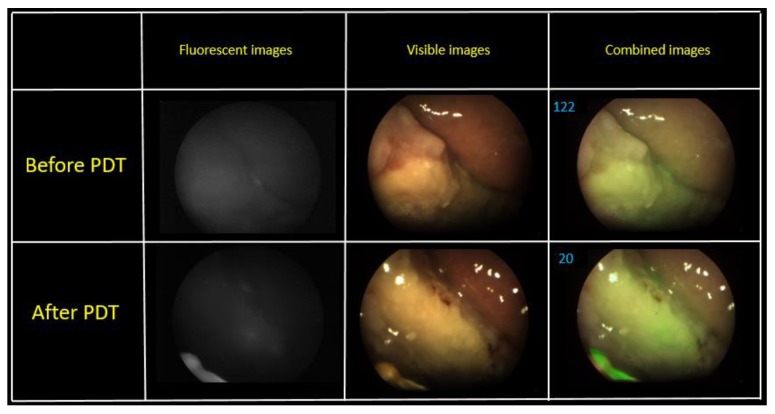
The combined images of mucous and tongue surface before and after PDT.

**Figure 15 jcm-08-02229-f015:**
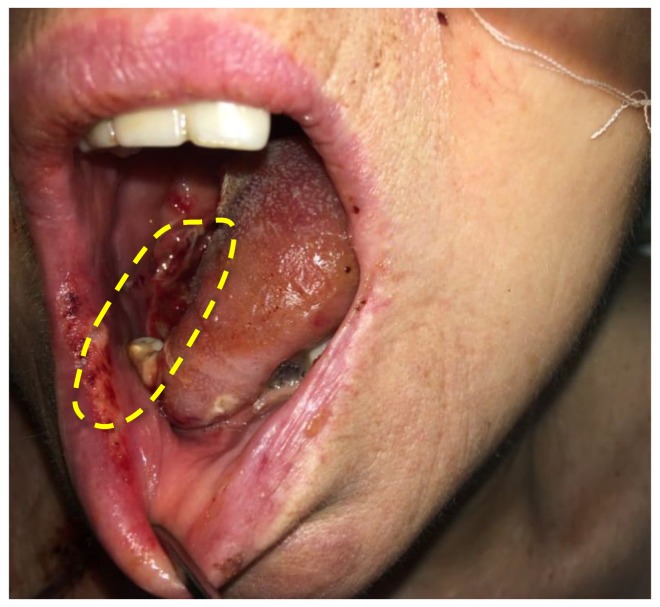
The photo of patient diagnosed with malignant neoplasms of the lateral tongue surface.

**Figure 16 jcm-08-02229-f016:**
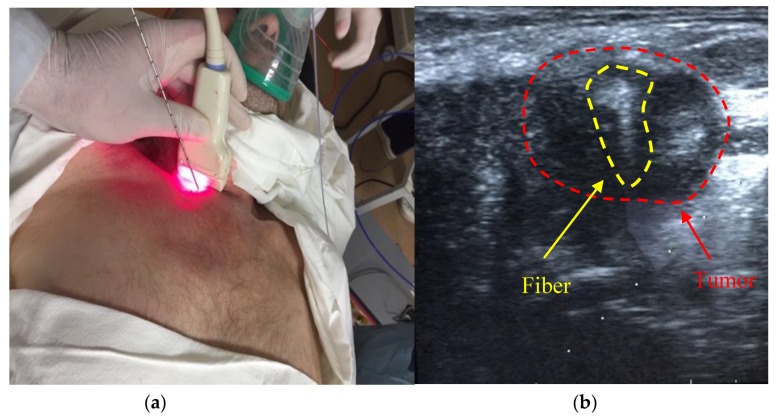
(**a**) The photo of stereotactic biopsy cannula introduction into the patient; (**b**) the sonogram of fiber introduction into the tumor area.

**Figure 17 jcm-08-02229-f017:**
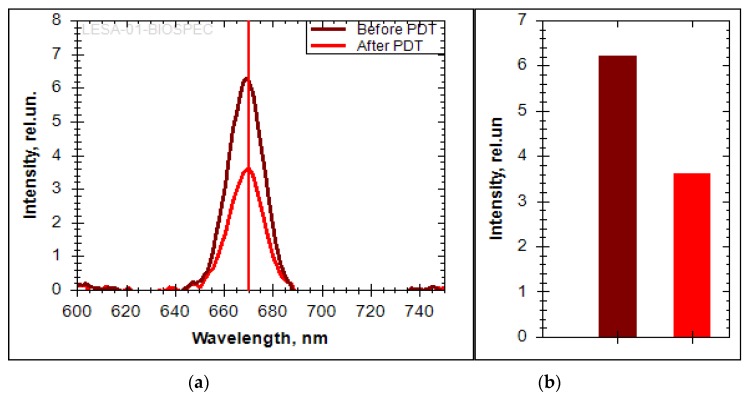
(**a**) The fluorescent spectra of Ce6 in the trachea; (**b**) the bar plot of Ce6 fluorescent intensity in the trachea.

**Figure 18 jcm-08-02229-f018:**
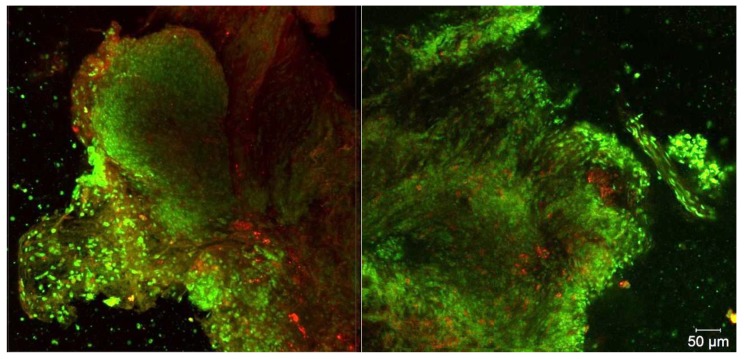
The 3D reconstruction of fluorescent images of biopsy after PDT. Green color corresponds to cells nuclei, red color corresponds to Ce6 fluorescence.

**Figure 19 jcm-08-02229-f019:**
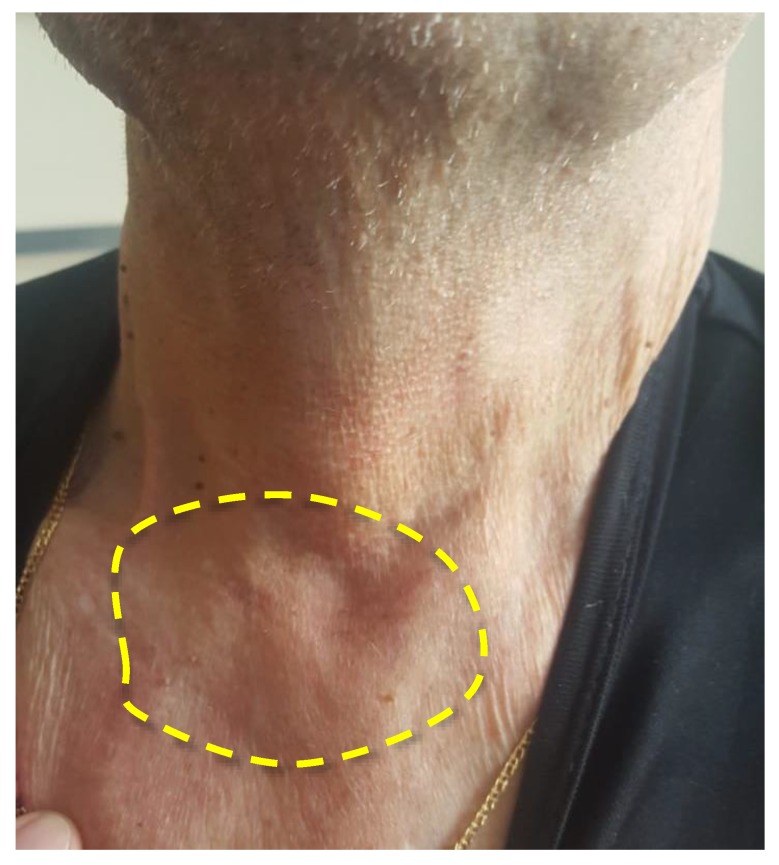
The photo of patient diagnosed with malignant neoplasm of the trachea with spread to the soft tissues in month after PDT.

**Figure 20 jcm-08-02229-f020:**
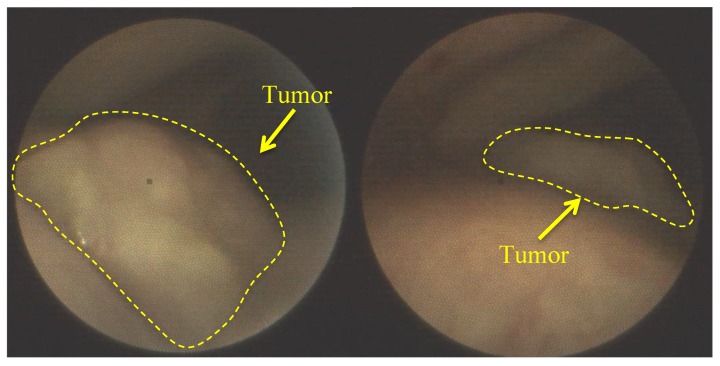
The visible images of vocal cord before PDT.

**Figure 21 jcm-08-02229-f021:**
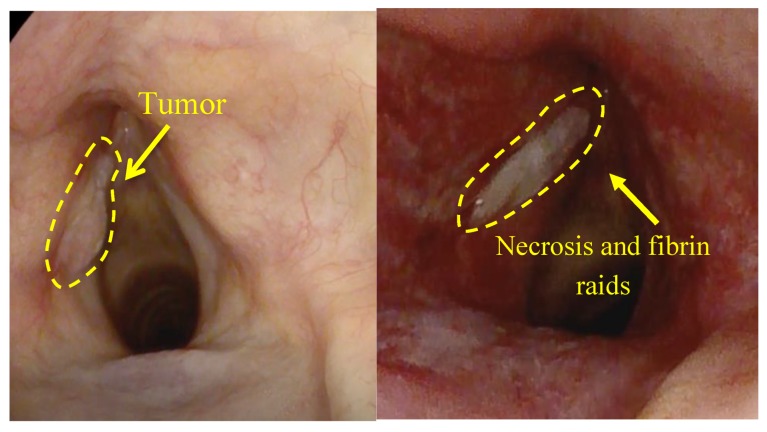
The visible images of vocal cord after PDT.

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
