# Peer review of "Trials of a Fluorescent Endoscopic Video System for Diagnosis and Treatment of the Head and Neck Cancer"

_jcm, 2019, doi:10.3390/jcm8122229_

Round 1

Reviewer 1 Report

As a clinician who utilizes PDT I am pleased to see the efforts to use fluorescence to guide PDT. However I think the manuscript needs some work to make it intelligable. As it is it reads like a roller coaster ride and I cannot really come to conclusions from the descriptions in the text. The patient group is very heterogenous so a statistical analysis is not feasible. I would recommend that the authors make this article into a case series and try to concentrate on describing fluorescence (quantitative or descriptive) tumor to normal tissue, and the quantitative photobleaching as it correaltes to clinical results. The discussion has to be expanded to explain the limitations of this study such as the limited number and variotions in locations and characters of the tumors.

Introduction:

The authors try to cram too much information into the introduction. The information about metastasis of the HNC described between lines 44-47 is unnecessary and also incomplete. I would reccomend removing this information.

The methods mentioned between lines 60-62 are experimental and not used routinely. The text gives the impression that theay are used routinely. I recommend to revise this sentence.

Methods:

It is not very structurally organized. The methods of measuring fluorescence is not explained. I am guessing that the evaluation is visual and not quantitative. This should be explicitely written. 

In the introduction the motivation is Photodiagnosis, however there is no description of how the authors have determined normal to tumor fluorescence ratios.

Results:

This section is also poorly organized. there is no quantitative data presented. the presentation is more like a case series. Whisch is perfectly fine but then the article should be organized differently. rather than methods and results the cases can be described separately.

Figure 16 is a photo of a cancer of the buccal mucosa not a tongue cancer!

There is no correlation with photobleaching and clinical results. 

I beleive if the authors want to correlate photobleaching with the effectiveness of PDT they should also be reporting the clinical resuolts and try to interpret these based on the fluorescence.

Discussion:

Too short. There is no explanations of the limitations of this study and what direction the future efforts should be heading.

Author Response

Thank you for your comments, we agree with all comments.

The patient group is very heterogenous so a statistical analysis is not feasible. I would recommend that the authors make this article into a case series and try to concentrate on describing fluorescence (quantitative or descriptive) tumor to normal tissue, and the quantitative photobleaching as it correaltes to clinical results.

Yes, we agree. The manuscript presents the initial results of malignant tumor diagnosis and treatment of two anatomical areas of the human body (head and neck). The main idea was to demonstrate a novel approach of intraoperative method of diagnosis and treatment control of hard-to-reach malignant tumors of different parts of head and neck via novel fluorescent endoscopic equipment in real time regime.

The discussion has to be expanded to explain the limitations of this study such as the limited number and variotions in locations and characters of the tumors.

The location and nature of the tumor formation included in this studied group were based on several criteria: 1) inconvenient location of the tumor for standard surgical treatment, which would require radical surgery, significantly reducing the quality of patient life; 2) a limited number of surgical equipment thus preventing an effective treatment; 3) the high recurrence rates.

Introduction: The authors try to cram too much information into the introduction. The information about metastasis of the HNC described between lines 44-47 is unnecessary and also incomplete. I would reccomend removing this information.

We agree.

deleted text:

Over the past five years, the survival rate was 63% of patients with this diagnosis. Survival of patients decreases with metastasis to the lungs and bones. The survival is only 6.6% with presence metastases in the upper gum region,and 4.1% is in the case of tongue carcinoma.In addition, there is evidence about the patients with metastases to the lymph nodes of the neck had tongue cancer (69.6%)

The methods mentioned between lines 60-62 are experimental and not used routinely. The text gives the impression that theay are used routinely. I recommend to revise this sentence.

The sentence was revised.

New sentence (lines 56-58):

However, the more exact methods as radio frequency spectroscopy, Raman spectroscopy, photoacoustics and optical coherence tomography are beginning to be applied more and more in oncology.

It is not very structurally organized. The methods of measuring fluorescence is not explained. I am guessing that the evaluation is visual and not quantitative. This should be explicitely written. 

The following paragraphs were inserted into the manuscript:

Lines (93-107)

The images in visible and fluorescent light in the areas of interest were obtained via video fluorescent system. In this case, the ultimate conclusion on the tumor process can’t be derived based on the image in visible light. Video fluorescence staining significantly increases the accuracy of visualization. Besides, a diagnostic marker is installed on the image fragment of the studied area allowing to measure the fluorescence on the selected area in real time. For determining the fluorescence index the software of the video fluorescence system calculates the average pixel intensity of the image obtained from the monochrome video camera in the highlighted area by the diagnostic cursor and normalizes it to the parameters of the monochrome video camera. Further, the average pixel intensity of the image of the color video camera red channel in the area highlighted by the diagnostic cursor is calculated and normalized to the parameters of the color video camera. Then, the first obtained value is normalized to the second one, thus obtaining the relative concentration of PS in the pathological tissue. The value of the normal tissue was determined by fluorescence and the degree of PS accumulation at two different points in the unchanged part of the studied organ. Identical values were obtained in these two positions, allowed to considering as a normal reference value [12].

 (lines 146-150)

FD of the pathological site was performed before and immediately after PDT procedure. Thе PS has an affinity to the tumor type, which considered in this study. The drug accumulates sufficiently in the tumor tissue, which allows identifying the difference the fluorescent signal in healthy tissue in comparison with pathological tissue. In addition, the drug accumulation depends on the time of diagnosis after PS administration.

In the introduction the motivation is Photodiagnosis, however there is no description of how the authors have determined normal to tumor fluorescence ratios.

The answer to this remark was inserted in the previous paragraph.

Results. This section is also poorly organized. there is no quantitative data presented. the presentation is more like a case series. Whisch is perfectly fine but then the article should be organized differently. rather than methods and results the cases can be described separately.

In the manuscript the patient histories were expanded and numerical results of the fluorescence index were inserted.

Figure 16 is a photo of a cancer of the buccal mucosa not a tongue cancer!

The patient diagnosed with cancer of the lateral surface of the tongue. PDT was carried out at the location of the malignant neoplasm and there was a photodynamic effect on the cheek part. However, a photo was taken where the location of tumor was in the shade, so another photo was inserted.

There is no correlation with photobleaching and clinical results.

Lines 340-345

The spectroscopy research of pathological part after PDT procedure showed the reduction of Ce6 fluorescent signal. The decrease of Ce6 fluorescent signal after PDT within the lesion area is explained by the partial destruction of PS molecules and a partial PS transformation into other forms. An exact value of the PS’s fluorescence intensity decrease cannot be calculated, because this value depends on the content of oxygen molecules and the PS localization in tumor cells (membrane, cytoplasm and organelles).

I beleive if the authors want to correlate photobleaching with the effectiveness of PDT they should also be reporting the clinical resuolts and try to interpret these based on the fluorescence.

We agree. The answer to this comment was inserted in the previous paragraph.

Discussion: Too short. There is no explanations of the limitations of this study and what direction the future efforts should be heading.

The discussion was expanded.

Reviewer 2 Report

The authors report their experience about intraoperative fluorescence imaging which provide a reliable information about borders of resected tumors. This method may ensure a quantitative approach of the analysis of tissue type during a surgery approach by accumulation of exogenous fluorophores in cancer cells. Selective accumulation of photosensitizers (PS) in malignant cells makes it possible to identify the boundaries and extent of the cancerous lesions by the fluorescence characteristic.

Then, the method of intraoperative fluorescent imaging should allow the determination of the exact location of the tumor and its boundaries. However, the possibility to update the technique by utilizing fluorescent antibodies in order to detect  stem cells markers, such as CD44, CD166 and CD133, as well as  VEGF to evaluate the increase of tumoral microvessels  to better identify the tumors should be discussed. Moreover, additional fluorescent probes to detect novel  molecular biomarkers at confocal microscopy,  should be mentioned as they could help to identify tumoral cells suggesting targeted therapies which can aid the patients after the therapeutic surgical approach.

The authors present  the convenience of endoscopic fluorescent  system in two nosologies of head and neck cancer. The authors may also discuss the use of this technique in different tumor district.

Author Response

Thank you for your comments, we agree with all comments.

The method of intraoperative fluorescent imaging should allow the determination of the exact location of the tumor and its boundaries. However, the possibility to update the technique by utilizing fluorescent antibodies in order to detect stem cells markers, such as CD44, CD166 and CD133, as well as  VEGF to evaluate the increase of tumoral microvessels  to better identify the tumors should be discussed. 

We agree. However, it was not the objectives of this study, but in further experiments to improve the methods of diagnosis of malignant tumors, our group suggests working with fluorescent antibodies.

Moreover, additional fluorescent probes to detect novel  molecular biomarkers at confocal microscopy,  should be mentioned as they could help to identify tumoral cells suggesting targeted therapies which can aid the patients after the therapeutic surgical approach.

We agree. In this study, the emphasis was placed on the intraoperative system of diagnosis and control of PDT by non-invasive method in real time mode, as another technique that allows arguing convincingly about the malignant process and conduct local antitumor therapy. We agree that the confocal microscopy study provides accurate results of the localization of cancer cells and will make it possible to make targeted therapy.

The authors present the convenience of endoscopic fluorescent  system in two nosologies of head and neck cancer. The authors may also discuss the use of this technique in different tumor district.

We agree.

The location and nature of the tumor formation included in this studied group were based on several criteria: 1) inconvenient location of the tumor for standard surgical treatment, which would require radical surgery, significantly reducing the quality of patient life; 2) a limited number of surgical equipment thus preventing an effective treatment; 3) the high recurrence rates.

.

Round 2

Reviewer 1 Report

The design of the manuscript is still problematic. However the content is an important contribution to the existing literature. With this heterogeneous patient group a better presentation does not seem feasible. Therefore I agree to approve th publication of this article. I am expecting a more structured and methodologically sound contributions from this group in future.